# Thyroid Dysfunction among Hypertensive Pregnant Women in Warri, Delta State, Nigeria

**DOI:** 10.3390/medicines9040029

**Published:** 2022-04-07

**Authors:** Philomena Nwabudike, Mathias Abiodun Emokpae

**Affiliations:** Department of Medical Laboratory Science, School of Basic Medical Sciences, University of Benin, Benin City 300283, Nigeria; nwabudikephilomena@gmail.com

**Keywords:** thyroid, pregnant women, Nigeria

## Abstract

The hypertensive pregnant woman is at a higher risk of complications either before, during or after birth and the baby can be adversely affected leading to preterm birth, low birth weight, placental separation (abruption) and other complications. The aim of the study was to evaluate thyroid dysfunction among pregnant women with hypertension. The study participants were 150 hypertensive pregnant women, 25 non-hypertensive pregnant women and 25 non-hypertensive non-pregnant women. Exactly 5mL of blood was collected and used for the assay of triiodothyronine (T3), thyroxine (T4) and thyroid-stimulating hormone (TSH) using the enzyme-linked immunosorbent assay technique. Data were analyzed using appropriate statistical tools. The results showed a significantly higher (*p* < 0.05) age among hypertensive pregnant women when compared with non-hypertensive pregnant women and non-hypertensive non-pregnant women. The serum TSH was significantly higher (*p* < 0.035) among hypertensive pregnant women when compared with non-hypertensive pregnant women. The triiodothyronine (T3) of hypertensive pregnant women was observed to be significantly higher (*p* < 0.05) when compared with both non-hypertensive pregnant women and non-hypertensive non-pregnant women. Some 15/150 (10%) of hypertensive pregnant women had subclinical hypothyroidism, 13/150 (8.7%) had overt hypothyroidism, while 122/150 (81.3%) were euthyroid. Among those with thyroid dysfunction, five and four of the subjects had subclinical hypothyroidism and overt hypothyroidism during the second trimester, while ten and nine had subclinical hypothyroidism and overt hypothyroidism during the third trimester, respectively. Evaluation of hypertensive pregnant women for thyroid function may be routinely performed to enable early diagnosis and treatment.

## 1. Introduction

Hypertension is a common medical problem affecting about 10% of pregnant women and may be accompanied by a high rate of fetal and maternal hospitalization and death [1,2]. Hypertensive disorders of pregnancy may represent different types of diseases, including chronic and gestational hypertension to eclampsia [1]. According to the National High Blood Pressure Education Program Working Group on High Blood Pressure in Pregnancy, hypertension in pregnancy can be sub-divided into four categories: (1) chronic hypertension, (2) preeclampsia-eclampsia, (3) preeclampsia superimposed on chronic hypertension and (4) gestational hypertension [3]. Despite the huge burden on maternal and neonatal health, evidence on the risk of thyroid dysfunction is limited. Hypertension is a condition in which the force of the blood against the artery walls is too high and may lead to organ damage and severe illnesses. Hypertension, which is defined as blood pressure ≥ 140/90 mmHg, has been recognized as one of the commonest chronic diseases [4]. High blood pressure during pregnancy has been reported to have several effects on the body and can put the mother at higher risk of complications before, during or after birth. It can affect the development of the placenta, resulting in an inadequate supply of nutrients and oxygen to the baby [5]. The prevalence of hypertensive disorders during pregnancy ranges from 10 to 21% among pregnant women. A prevalence of 21.6% was reported in south-eastern Nigeria [6], 10% in Ibadan [7], 11.6% in Benin City [8] and 17% in Sokoto [9].

Thyroid dysfunctions (hyperthyroidism and hypothyroidism) during pregnancy may precipitate hypertension and other associated adverse health outcomes for both fetus and mother. These adverse health outcomes include increased risk of miscarriage, pregnancy induced hypertension, preterm delivery, placental abruption, low birth weight, and fetal death [10]. Additionally, hypothyroidism may be an independent risk factor for preeclampsia and fetal growth restriction. However, the underlying roles of thyroid dysfunction in hypertensive disorders during pregnancy are not completely understood [11].

It is of public health importance to know the thyroid hormone abnormalities associated with hypertension in pregnancy in Delta State of Nigeria. This may help to assist in the management of pregnant women with hypertension. Normal levels of maternal thyroid hormones play important roles in maintaining systemic homeostasis and improving pregnancy outcomes [12,13]. The aim of the study was to evaluate thyroid dysfunction among pregnant women with hypertension attending clinics in Warri, Delta State, Nigeria.

## 2. Materials and Methods

### 2.1. Study Population and Design

This is a cross-sectional study of hypertensive pregnant women at different trimesters attending antenatal care clinics in some primary health centers in Warri (Pessu and Eboh Health Centers). The study population was stratified according to the age of pregnancy (trimesters): Pregnant women in the first trimester with hypertension, pregnant women in the second trimester with hypertension, pregnant women in the third trimester with hypertension, pregnant women without hypertension as control subjects 1 and non-pregnant women as control subjects 2.

The locally established reference ranges are:TSH = 0.50–4.1 µIU/mL;T3 = 0.52–1.85 ng/dL;T4 = 4.8–11.6 µg/dL.

The thyroid dysfunctions were defined based on the locally established reference ranges as follows:Hypothyroidism—TSH > 4.1 µIU/mL with T4 < 4.8 µg/dL;Subclinical hypothyroidism—TSH > 4.1 µIU/mL with normal T4 and T3;Hyperthyroidism—TSH < 0.5 µIU/mL with T4 > 11.6 µg/dL.

### 2.2. Ethical Consideration

Ethical approval was sought and obtained from the Ethical Committees of Delta State Ministry of Health, Delta State (Reference HM/596/T/103 dated 7th September 2020). Consent was obtained from all the participants before the commencement of sample collection. 

### 2.3. Inclusion and Exclusion Criteria

Hypertensive pregnant women attending antenatal clinics at the health centers without diabetes or other chronic illnesses were included in the study. The diagnostic criterion for hypertension in pregnancy was an elevated blood pressure to ≥140/90 mmHg in previously normotensive women. Non-hypertensive pregnant women and non-pregnant women were added as control subjects. Pregnant women with multiple pregnancies, known thyroid disorders, chronic hypertension on medication and gestational diabetes mellitus were excluded from the study. Additionally, those who were smokers, those living with HIV infection, pulmonary tuberculosis, mental illness, hepatitis and syphilis were also excluded from the study.

### 2.4. Sample Size Determination

The minimum sample size was determined using the sample size determination formula for health studies [14]. A prevalence of 11.7% of hypertension among pregnant women in Benin City [8] was used for this study.
*N* = Z^2^ × P (1 − P)/d^2^
where *n* = minimum sample size, P = estimated prevalence, Z = standard normal deviate that corresponds to 95% confidence limit (1.96) and 1 is the alpha level of significance (5%).

Using this formula, the calculated sample size was 149, and 150 hypertensive pregnant women, 25 euthyroid pregnant women and 25 non-pregnant women were also recruited for this study.

### 2.5. Anthropometric and Demographic Measurements

The weight, height and blood pressure of every participant were measured with the help of a research assistant. The body mass index (BMI) was obtained by calculations.
Body Mass Index (BMI) = Weight (kg)/height (m^2^).

A semi-structured questionnaire was administered to obtain the socio-demographic characteristics of the participants.

### 2.6. Sample Collection

Using aseptic precaution, 5 mL of venous blood was collected into a plain container and allowed to clot for 15 min. The clotted blood sample was centrifuged at 3000 rpm for 10 min, and the serum was separated into cryovial and kept frozen at −20 °C until analyzed.

### 2.7. Biochemical Analysis

The thyroid hormones were determined using the enzyme-linked immunosorbent assay (ELISA). The laboratory analyses were carried out in Central Hospital, Warri, using the ELISA method.

#### Principle

The principle for Thyroxine (T4) and Triidothyronine (T3) test system is shown below. 

The reagents needed for a solid-phase enzyme immunoassay include an immobilized antibody, enzyme-antigen conjugate and native antigen. Upon mixing the immobilized antibody, enzyme-antigen conjugate and a serum containing the native antigen, a competition reaction results between the native antigen and the enzyme–antigen conjugate of a limited number of insolubilized bind sides.

The interaction is illustrated by the equation below:En3Ag + Ag + Abcw _k−a_ ⇄ ^ka^ AgAcw + En3AgAbcw

After equilibrium is attained, the antibody-bound fraction is separated from the unbound antigen by decantation or aspiration. The enzyme activity in the antibody-bound fraction is inversely proportional to the native antigen concentration. By using several different serum references of known antigen concentrations, a dose–response curve can be generated, from which the antigen concentration of an unknown can be ascertained.

The principle for Thyrotropin (TSH), which is slightly different, is as follows; the essential reagents required for an immunoenzymometric assay include high affinity and specificity antibodies (enzyme-conjugated and immobilized) with different and distinct recognition and native antigen. In this procedure, the immobilization takes place during the assay at the surface of a microplate well through the interaction of streptavidin coated on the well and exogenously added biotinylated monoclonal anti-TSH antibody.

Upon mixing the monoclonal biotinylated antibody, the enzyme labeled antibody and serum containing the native antigen, a reaction results between the native antigen and the antibodies without the competition of hindrance to form a soluble sandwich complex. The interaction is illustrated by the following equation.
EnzAb + Ag _(TSH)_ + Ab _k−a_ ⇄ ^ka^ EnzAb − Ag_TSH_ − Ab

Simultaneously, the complex is deposited to the well through the high affinity reaction of streptavidin and biotinylated antibody, as illustrated below.
Ab − Ag _(TSH)_ − Ab + streptavidin → immobilized complex

Streptavidin = Streptavidin immobilized on well immobilized complex = sandwich complex bound to the well surface.

After equilibrium is attained, the antibody-bound fraction is separated from the unbound antigen by decantation or aspiration. The enzyme activity in the antibody-bound fraction is directly proportion to the native antigen concentration.

A set of calibrators were assayed to ensure linearity, commercial quality control sera were included in the runs, samples were determined in duplicates and mean values were calculated.

### 2.8. Statistical Analysis

Data generated were analyzed using Statistical Package for the Social Sciences software (v20.0, IBM SPSS, Armonk, NY, USA). Comparison between means was conducted using the Student *t*-test and ANOVA (Analysis of variance), and the Pearson correlation coefficient was used to determine the association between the measured variables and clinical condition. The level of significance was set at a *p*-value of 0.05.

## 3. Results

A total of 200 women volunteered, which consisted of 150 hypertensive pregnant women, 25 non-hypertensive pregnant women and 25 non-hypertensive non-pregnant women were recruited for the study. 

The results of this study show a significantly higher (*p* < 0.001) age in hypertensive pregnant women when compared with non-hypertensive pregnant women and non-hypertensive non-pregnant women. However, there was no significant difference (*p* > 0.05) observed when the age of non-hypertensive pregnant women and non-hypertensive non-pregnant women were compared. Likewise, the weight of the hypertensive pregnant women was significantly higher (*p* < 0.001) when compared with non-hypertensive pregnant women and non-hypertensive non-pregnant women. However, no significant difference (*p* > 0.05) was observed when the weight of non-hypertensive pregnant women and non-hypertensive non-pregnant women were compared, as shown in Table 1 below. There was no significant difference (*p* > 0.05) observed in the height of all the respondents when compared. The systolic blood pressure of hypertensive pregnant women was observed to be significantly higher (*p* < 0.001) when compared with non-hypertensive pregnant women and non-hypertensive non-pregnant women, respectively, but no significant difference (*p* > 0.05) was observed when the systolic blood pressure of non-hypertensive pregnant women and non-hypertensive non-pregnant women was compared. Likewise, diastolic blood pressure was significantly higher (*p* < 0.001) among hypertensive pregnant women than in non-hypertensive pregnant women and non-hypertensive non-pregnant women. Additionally, diastolic blood pressure was significantly higher (*p* < 0.001) among non-hypertensive pregnant women than in non-hypertensive non-pregnant women (Table 1).

The serum TSH was significantly higher (*p* < 0.035) among hypertensive pregnant women when compared with non-hypertensive pregnant women. On the other hand, there was no significant difference (*p* > 0.05) in TSH among hypertensive pregnant women compared with non-hypertensive non-pregnant women. Additionally, TSH was significantly lower (*p* < 0.05) among non-hypertensive pregnant women than in non-hypertensive non-pregnant women. The mean triiodothyronine (T3) among hypertensive pregnant women was observed to be significantly higher (*p* < 0.001) than both non-hypertensive pregnant women and non-hypertensive non-pregnant women. However, there was no significant difference (*p* > 0.05) when the T3 of both non-hypertensive pregnant and non-hypertensive non-pregnant women were compared, and no significant difference (*p* > 0.05) was also observed in thyroxine (T4) levels when hypertensive pregnant women and non-hypertensive pregnant women were compared. On the other hand, the mean serum T4 was significantly lower (*p* < 0.05) in both hypertensive pregnant women and non-hypertensive pregnant women than in non-hypertensive non-pregnant women (Table 2).

There was no significant difference (*p* > 0.05) observed in TSH and T3 of the individuals in the first trimester, but during the second and third trimesters, significantly higher (*p* < 0.05) levels of T3 were observed among non-hypertensive pregnant women more than non-hypertensive non-pregnant women. There was a significant difference (*p* < 0.05) observed between the T4 of the first trimester compared with the second and third trimesters. However, the T4 of the first trimester was observed to be significantly lower (*p* < 0.05) than the T4 of non-hypertensive non-pregnant women when compared. The second trimester had a significantly higher (*p* < 0.05) T4 than the third trimester and significantly lower (*p* < 0.05) T4 than non-hypertensive non-pregnant women when compared. Likewise, the third trimester shows significantly lower (*p* < 0.05) T4 than non-hypertensive non-pregnant women (Table 3).

The comparison of thyroid hormone levels between hypertensive pregnant women and non-hypertensive pregnant women indicates that the TSH level was significantly lower (*p* < 0.009) than non-hypertensive pregnant women. Additionally, the mean T4 level among hypertensive pregnant women was significantly lower (*p* < 0.001) than in non-hypertensive pregnant women. There was no significant difference (*p* > 0.05) in the mean levels of T3 between hypertensive and non-hypertensive pregnant women (Table 4).

To better determine the possible impact of thyroid dysfunction on blood pressure, the thyroid hormone levels of hypertensive pregnant women whose blood pressure was >140/90 mmHg were compared with those whose blood pressure was <140/90 mmHg. The mean T3 of those with blood pressure >140/90 mmHg was significantly lower (*p* < 0.001) than those with blood pressure <140/90 mmHg. Conversely, the mean TSH of those with blood pressure >140/90 mmHg was significantly lower (*p* < 0.05) than those of pregnant women with blood pressure <140/90 mmHg. No significant difference in the mean T4 was observed between the two groups (Table 5).

Table 6 shows the distribution of thyroid dysfunction among hypertensive pregnant women. Some 15/150 (10%) of them had subclinical hypothyroidism, 13/150 (8.7%) had overt hypothyroidism, while 122/150 (81.3%) were euthyroid. Among those with thyroid dysfunction, five and four of the subjects had subclinical hypothyroidism and overt hypothyroidism during the second trimester, while ten and nine had subclinical hypothyroidism and overt hypothyroidism during the third trimester, respectively (Table 7).

## 4. Discussion

Maternal thyroid hormone circulating levels play very important roles in maintaining systemic homeostasis and have been reported to be associated with many health-related effects in pregnant women and their unborn children [15]. Several authors have reported that thyroid dysfunction is associated with hypertension and hypertensive disorders among pregnant women [2,13]. Changes in thyroid function can affect several organs system of the body and may lead to hypertensive complications in pregnancy [16]. Physiological alterations in thyroid hormones occur during pregnancy. Pregnancy-related hormones (human chorionic gonadotropin (hCG) and estrogen can cause increased levels of T3 and T4. Human chorionic gonadotropin does stimulate the thyroid gland mimicking TSH action, thereby suppressing TSH production, while estrogen stimulates the production of thyroid-binding globulin (the transporter of thyroid hormones). These physiologic actions may cause T3 and T4 to increase during the first trimester until about the 16 weeks of gestation and then stabilize. This can sometimes make thyroid hormone tests difficult to interpret. It is of public health importance to know the thyroid hormone abnormalities associated with hypertension in pregnancy in our setting. This may help to assist in the management of pregnant women with hypertension. Normal levels of maternal thyroid hormones enhance systemic homeostasis and improve pregnancy outcomes [12,13].

The anthropometric variables of the study participants were similar because they were women of child-bearing age. However, there was a significantly higher (*p* < 0.05) age in hypertensive pregnant women compared with non-hypertensive pregnant women and non-hypertensive non-pregnant women. However, there was no significant difference (*p* > 0.05) observed when the age of non-hypertensive pregnant women was compared with that of non-hypertensive non-pregnant women. This is in agreement with previous authors in a study conducted among subjects with gestational hypertension in Benin, South–South Nigeria [17]. Increased maternal age may be a risk factor for the development of hypertension in pregnancy. The pathway of this association is not well understood but might be due to changes in vascular endothelial as a result of ageing. Other confounding factors, such as parity and body mass index, might explain the association. During early pregnancy, the flow of blood within the circulatory blood vessels and mean arterial blood pressure decline and the cardiac output increases [18]. In an earlier study conducted among pregnant Nigerian women, with a mean age of 28 years, it was reported that maternal age was not associated with systolic blood pressure in any trimester [19]. A positive correlation was reported between maternal age and diastolic blood pressure at 30–38 weeks gestation [19]. However, Gaillard et al. [18] reported an association between older maternal age and higher third-trimester diastolic blood pressure. The authors concluded that the blood pressure differences between younger and older women appear to be small, and the results regarding the association of maternal age with the risk of gestational hypertensive disorders are inconsistent. A large Swedish population-based cohort study among nulliparous women aged ≤34 years reported that maternal age was not associated with hypertension in pregnancy [20]. There was a positive correlation between age and hypertension in this study. Additionally, the weight of the hypertensive pregnant women was significantly higher when compared with non-hypertensive pregnant women and non-hypertensive non-pregnant women. However, no significant difference was observed when the weight of non-hypertensive pregnant women and non-hypertensive non-pregnant women were compared. Adebayo et al. [21] reported in their work that weight is a risk factor for hypertension, but Ogah et al. [22], in their work, also observed that body mass index (BMI) is a risk factor for hypertension. The relationship between Blood pressure and body fat is not restricted to the morbidly obese but is continuous throughout the entire range of body weight. The mechanism by which obesity raises blood pressure is not fully understood, but increased BMI is associated with an increase in plasma volume and cardiac output [23]. Because excessive weight gain during pregnancy has been associated with the risk of developing hypertension, entering pregnancy with a healthier BMI has better immediate and long-term outcomes. Some authors have shown that the risk of gestational hypertension correlated positively with BMI, with a relative risk of 2.24 for a BMI increase of 3 kg/m^2^ [23].

This study observed a significantly higher (*p* < 0.05) TSH in hypertensive pregnant women when compared with non-hypertensive pregnant women, but no significant difference (*p* > 0.05) was observed when compared with non-hypertensive non-pregnant women. However, there was a significantly higher (*p* < 0.05) TSH in non-hypertensive pregnant women when compared with non-hypertensive non-pregnant women. This is in tandem with a previous study among hypertensive pregnant women in Kano, Nigeria [24]. There was significantly higher (*p* < 0.05) T_3_ in hypertensive pregnant women when compared with non-hypertensive pregnant and non-hypertensive and non-pregnant women. However, there was no significant difference (*p* > 0.05) observed when non-hypertensive pregnant and non-hypertensive non-pregnant women were compared. This report is at variance with an earlier report by Abdulslam and Yahaya [24] that observed no differences between gestational hypertension and non-hypertensive. The differences in the observation may be due to the fact that this study evaluated total T_3_, whereas Abdulslam and Yahaya [24] evaluated free T_3_. There was a positive correlation between systolic blood pressure and T_3_. This finding is in agreement with a previous study [25]. Pregnancy impacts the functioning of the thyroid gland and is associated with an increase in the size of the gland (iodine-replete areas show a greater increase), thereby leading to an increase in the production of triiodothyronine (T3). Additionally, human chorionic gonadotrophin (hCG) secreted by the placenta is also reported to impact thyroid function [26].

This study indicated that 10% of hypertensive pregnant women had subclinical hypothyroidism, 8.7% had overt hypothyroidism and 81.3% were euthyroid. Thyroid dysfunction is common among pregnant women. Earlier reports elsewhere showed that the prevalence of clinical overt hyperthyroidism or subclinical hyperthyroidism was about 0.1–0.4% during pregnancy. The prevalence of hypothyroidism was about 2.5%, clinical hypothyroidism was 0.2–0.3% and subclinical hypothyroidism was observed to be 2–3% [27]. A study in a general population in China reported a prevalence of gestational clinical hypothyroidism of 0.5–0.6% and subclinical hypothyroidism of 2–3%. Subclinical hypothyroidism was observed to be the highest [28].

Hypothyroidism is a known cause of hypertension. If untreated, subclinical hypothyroidism and overt hypothyroidism might lead to several adverse outcomes in the mother and fetus [27,29,30]. Some authors reported that even after treatment for hypothyroid diseases, these women might also be at increased risk for preeclampsia [31], a finding that was at variance with that of Casey et al. [32], who did not find any increased incidence of preeclampsia among women treated for subclinical hypothyroidism. 

Hypothyroidism has been reported to be associated with various vascular pathogenic effects, including endothelial cell dysfunction [27], which is also an abnormal physiological basis of gestational hypertension. In a large study of women with singleton births, hypothyroid mothers in Finland were reported “to be associated with higher risks of gestational hypertension (OR = 1.20, 95% CI 1.10–1.30), severe preeclampsia (OR = 1.38, 95% CI 1.15–1.65), preterm births (OR = 1.25, 95% CI 1.16–1.34), and neonatal intensive care unit admission (OR = 1.23, 95% CI 1.17–1.29)” [33]. Other authors observed that elevated maternal serum TSH (higher than 10 mIU/L) correlated with an increased risk of stillbirth [27]. The study by Casey et al. [32] involving 25,756 women showed that subclinical hypothyroidism in pregnancies was linked with a three-times increased risk of placental abruption. Additionally, the risk of preterm birth was about two times higher in women with subclinical hypothyroidism than in euthyroid pregnant women. In this study, only two types of thyroid dysfunction—subclinical hypothyroidism and overt hypothyroidism—were observed among hypertensive pregnant women. Lai et al. [34] observed that hypothyroidism, subclinical hypothyroidism and high TSH levels were associated with an increased risk of gestational hypertension. Although screening of pregnant women for thyroid dysfunction is not generally accepted, the proportion of hypertensive women with thyroid dysfunction is relatively high, indicating that early diagnosis and treatment will be necessary to improve pregnancy outcomes.

In this study, it was observed that hypothyroidism, subclinical hypothyroidism and high TSH levels were associated with an increased risk of hypertension in pregnant women. Some authors have shown that hyperthyroidism and high fT4 levels are risk factors for hypertensive disorders during pregnancy [35], but in this study, it was seen that none of the hypertensive pregnant women had hyperthyroidism. Additionally, overt and subclinical hypothyroidism was related to endothelial dysfunction, which was considered a cause of hypertension [36]. The adrenergic system is the major regulator of cardiac and vascular function, and this may be accomplished through the activation of specific receptors located on the endothelial surface by the local and systemic release of catecholamines [37]; therefore, the effects of thyroid hormones on hypertension may be through the regulation of the adrenergic system. Pregnant women with hypothyroidism could be at risk of obstetrical complications (including intrauterine fetal death, gestational hypertension, placental abruption and poor perinatal outcome). Evidence has shown that thyroid hormone administration significantly improves such complications. However, babies of hypothyroid mothers may be healthy without evidence of thyroid dysfunction [38]. Conversely, some authors have reported a risk of higher perinatal mortality, congenital malformation and low birth weight [39].

The limitations of this study are the small sample size and the fact that it was conducted in two major towns in Delta State. Therefore, a generalization about the Nigerian population cannot be made. Further studies involving a large sample size are suggested.

## 5. Conclusions

It was observed from this study that there were alterations in the thyroid function of pregnant women with hypertension. Some 10% of hypertensive pregnant women had subclinical hypothyroidism, 8.7% had overt hypothyroidism and 81.3% were euthyroid. Among those with thyroid dysfunction, five and four of the subjects had subclinical hypothyroidism and overt hypothyroidism during the second trimester, while ten and nine had subclinical hypothyroidism and overt hypothyroidism during the third trimester, respectively. An evaluation of hypertensive pregnant women for thyroid function may be routinely carried out to enable early diagnosis and treatment and to improve pregnancy outcomes.

## Figures and Tables

**Table 1 medicines-09-00029-t001:** Anthropometric measurements of hypertensive pregnant women, non-hypertensive pregnant women and non-hypertensive non-pregnant women.

Parameter	Hypertensive Pregnant Women = A (*n* = 150)	Non-Hypertensive Pregnant Women = B (*n* = 25)	Non-Hypertensive Non-Pregnant Women = C (*n* = 25)	*F* Value	Sig
Age (Years)	31.67 ± 0.44 (30.7–32.5)	25.88 ± 1.08 (23.6–28.1)	28.72 ± 1.21 (26.2–31.2)	13.623	0.001
Weight (Kg)	80.65 ± 1.09 (78.4–82.7)	68.48 ± 2.56 (63.2–73.7)	64.48 ± 1.85 (60.6–68.2)	23.411	0.001
Height (M)	161.27 ± 0.47 (160.3–162.2)	164.24 ± 1.39 (161.3–167.1)	162.68 ± 0.43 (159.7–165.6)	2.829	0.061
BMI	25.07 ± 0.33 (24.2–25.8)	20.95 ± 0.72 (19.8–21.2)	20.17 ± 0.49 (19.9–21.1)	26.185	0.001
SBP (mmHg)	150.72 ± 0.96 (148.8–152.6)	112.48 ± 1.72 (108.9–116.0)	114.16 ± 1.97 (110–118.2)	208.206	0.001
DBP (mmHg)	88.28 ± 0.66 (86.9–89.5)	62.44 ± 1.39 (59.5–65.3)	68.84 ± 1.77 (65.1–72.4)	152.596	0.001
Gestational age	23.90 ± 0.622	27.36 ± 1.15	0.00 ± 0.00	138.692	0.001
	A vs. B	A vs. C	B vs. C		
Age (Years)	0.001	0.014	0.068		
Weight (Kg)	0.001	0.001	0.272		
Height (M)	0.055	0.286	0.366		
BMI	0.001	0.001	0.473		
SBP (mmHg)	0.001	0.001	0.597		
DBP (mmHg)	0.001	0.001	0.005		
Gestational Age	0.022	0.001	0.001		

95% Confidence interval in parenthesis; BMI = body mass index; SBP = systolic blood pressure; DBP = diastolic blood pressure.

**Table 2 medicines-09-00029-t002:** Thyroid hormone levels among hypertensive pregnant women, non-hypertensive pregnant women and non-hypertensive non-pregnant women.

Parameters	Hypertensive Pregnant Women =A (*n* = 150)	Non-Hypertensive Pregnant Women = B (*n* = 25)	Non-Hypertensive Non-Pregnant Women = C (*n* = 25)	*F* Value	Sig
TSH (µIU/mL)	3.08 ± 0.09 (2.00–3.35)	1.66 ± 0.09 (1.47–1.84)	3.18 ± 0.35 (1.94–4.42)	4.681	0.010
T3 (ng/mL)	1.02 ± 0.06 (1.27–1.78)	0.99 ± 0.06 (0.86–1.12)	1.04 ± 0.08 (0.87–1.20)	8.084	0.001
T4 (µg/dL)	6.53 ± 0.85 (6.03–6.63)	6.34 ± 0.22 (5.89–6.78)	7.92 ± 0.36 (7.17–8.66)	8.676	0.001
Post Hoc	A vs. B	A vs. C	B vs. C		
TSH (µIU/mL)	0.035	0.062	0.003		
T3 (ng/mL)	0.002	0.005	0.782		
T4 (µg/dL)	0.993	0.001	0.002		

95% Confidence interval in parenthesis; TSH = Thyroid-stimulating hormone; T3 = triiodothyronine; T4 = Thyroxine.

**Table 3 medicines-09-00029-t003:** Serum thyroid hormone levels at first, second and third trimesters of pregnancy among hypertensive and non-hypertensive pregnant women compared with Non-Pregnant women.

Parameter	First Trimester = A (*n* = 18)	Second Trimester = B (*n* = 56)	Third Trimester = C (*n* = 101)	Non-Pregnant = D (*n* = 25)	*F* Value	Sig
TSH (µIU/mL	1.89 ± 0.61 (1.52–2.14)	2.36 ± 0.17 (2.01–2.64)	2.13 ± 0.12 (1.83–2.23)	3.18 ± 0.35 (1.94–4.42)	1.809	0.147
T3 (ng/dL)	1.33 ± 0.11 (1.12–1.55)	1.47 ± 0.05 (1.33–1.54)	1.48 ± 0.11 (1.10–1.85)	1.04 ± 0.08 (0.87–1.20)	2.645	0.061
T4 (µg/dL)	6.23 ± 0.43 (5.44–7.21)	6.74 ± 0.27 (6.18–7.21)	6.10 ± 0.20 (5.80–6.46)	7.90 ± 0.36 (7.17–8.66)	1.856	0.115
Post Hoc	A vs. B	A vs. C	A vs. D	B vs. C	B vs. D	C vs. D
TSH (µIU/mL	0.164	0.451	0.057	0.290	0.330	0.064
T3 (ng/mL)	0.471	0.427	0.207	0.953	0.008	0.015
T4 (µg/dL)	0.327	0.786	0.004	0.052	0.009	0.001

**Table 4 medicines-09-00029-t004:** Serum thyroid hormone levels among hypertensive pregnant women compared with non-hypertensive pregnant women.

Parameter	Hypertensive Pregnant Women (*n* = 150)	Non-Hypertensive Pregnant Women (*n* = 25)	*t* Value	Sig
TSH (µIU/mL)	3.08 ± 0.09	2.64 ± 0.35	−2.732	0.009
T3 (µIU/mL)	1.02 ± 0.85	1.64 ± 0.08	−0.521	0.605
T4 (µIU/mL)	6.53 ± 0.85	7.92 ± 0.36	−3.768	0.001

**Table 5 medicines-09-00029-t005:** Comparison of thyroid hormone levels among hypertensive pregnant women with Blood pressure greater than 140/90 mmHg and less than 140/90 mmHg.

Variables	Blood Pressure >140/90 mmHg (*n* = 139)	Blood Pressure <140/90 mmHg (*n* = 36)	*p*-Value
T3 (ng/mL)	1.37 ± 0.42 (1.30–1.44)	1.75 ± 0.53 (0.68–2.81)	0.001
T4 (µg/dL)	6.32 ± 1.85 (6.01–6.63)	6.36 ± 1.42 (5.88–6.84)	0.86
TSH (µIU/mL)	2.12 ± 1.96 (1.95–2.29)	2.42 ± 1.11 (1.65–2.40)	0.05

95% Confidence interval in parenthesis; TSH = Thyroid-stimulating hormone; T3 = triiodothyronine; T4 = Thyroxine.

**Table 6 medicines-09-00029-t006:** Distribution of thyroid dysfunction among hypertensive pregnant women.

Thyroid Function Status	Number of Subjects (%)	T3 levels (Ref Range 0.52–1.85)	T4 Levels (Ref Range 4.8–11.6)	TSH Levels (Ref Range 0.5–4.1)
Euthyroid	122 (81.3)	0.98 ± 0.21	6.24 ± 1.60	2.62 ± 0.62
Subclinical hypothyroidism	15 (10)	1.01 ± 0.21	11.2 ± 0.81	4.81 ± 0.19
Overt Hypothyroidism	13 (8.7)	0.97 ± 0.72	4.1 ± 0.12	4.92 ± 0.16
Total	150 (100)	1.02 ± 0.06	6.53 ± 0.85	3.08 ± 0.09

Reference range in parenthesis; TSH = Thyroid-stimulating hormone; T3 = triiodothyronine; T4 = Thyroxine.

**Table 7 medicines-09-00029-t007:** Distribution of thyroid dysfunction according to gestational age among hypertensive pregnant women.

Thyroid Function Status	2nd Trimester (%)	3rd Trimester (%)	Total (%)
Euthyroid	38 (31.1)	84 (68.9)	122 (100)
Subclinical hypothyroid	05 (33.3)	10 (66.7)	15 (100)
Overt hypothyroidism	04 (30.8)	9 (69.2)	13 (100)

## Data Availability

Data from MSc project conducted at the Department of Medical Laboratory Science, University of Benin, Benin City, Nigeria.

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
