# Peer review of "Thyroid Dysfunction among Hypertensive Pregnant Women in Warri, Delta State, Nigeria"

_medicines, 2022, doi:10.3390/medicines9040029_

Round 1
Reviewer 1 Report
Thyroid dysfunction among hypertensive women in Warri, Delta State Nigeria.
Nawabudike and Emokpae
I enjoyed reading this paper and commend the authors for their work and efforts to bring it together. I always approach other authors’ works with the premise that I am going to recommend publication unless there are glaring errors. I fear that this study has a few of those - to which the authors may wish to give some thought.
I understand the equation that determined the sample size and the authors have applied it accurately, but the study size (150 hypertensive pregnant women) does seem low and the numbers in the control groups (25 in each) similarly look too low to make any valid conclusions from the data. Rather than settle for the minimum number as defined by this formula, it would have been better to have increased the sample size to make any conclusions the authors draw seem more convincing.
It is known that thyroid function changes throughout pregnancy e.g., increased thyroid binding globulin in pregnancy will cause rises in total T3 and total T4. Increased HCG in the first trimester will suppress TSH to start with and this effect disappears by week 16 where the TSH becomes more reliable as a measure of thyroid function. I’m not sure that the authors have included details like this in their work.
I would be interested to learn how “mental illness” is defined by the authors and why it became an exclusion factor in their study population. I understand how thyroid disease can cause depressive illness, anxiety etc but I’m much less familiar with a reverse causative relationship i.e., does depressive illness, anxiety etc cause thyroid dysfunction and thus exclude such patients from this study.
The assay principle is described in minute detail – as this is a non-commercial ELISA assay for all three analytes, this probably has some relevance, but I would suggest that most readers of this work will not be particularly interested in the finer details. However, as these results are not generated from commercial kits (and there is no criticism here), the authors really have to say whether these assays are subjected to an external QC scheme and that the performance of the assays are acceptable within that QC scheme. Otherwise, how are we meant to believe the significance of any results generated?
Moving onto the presentation of the results: I’m sorry to say that I didn’t learn anything that I didn’t know already. However, one of my major difficulties with these data is that as far as I can see, the authors don’t define any reference ranges for their results: perhaps they haven’t defined the reference ranges for their ELISA assays but without this, it is very difficult to draw any conclusions from their work.
Similarly, the authors say that a number of women had overt hypothyroidism and some had subclinical hypothyroidism, but they have not defined what they consider the TSH to be in such cases. I would have said that at TSH above 10mU/L would be considered to represent overt hypothyroidism and values above the top of the reference range but <10mU/L as defining “subclinical hypothyroidism”. As far as I can see (and remember, I don’t know what is considered a normal value in their assays) all the TSH values in Tables 2 and 6 for example, look completely normal! There may be statistically significant differences between the study groups but if all these differences lie within a normal range, I wonder what significance that has i.e., as a clinician, what do I do to intervene when I see a slightly higher TSH value that still lies within my reference range? Given how labile TSH can be, I would do nothing to alter my management. In Table 6, if I’m reading this correctly, the authors are suggesting a TSH of 0.41 and TSH of 0.32 represent subclinical and overt hypothyroidism respectively. The TSH values don’t suggest any form of hypothyroidism to me and in fact these TSH values are actually lower than the euthyroid value of 2.62 – that just doesn’t make sense.
It would have been more powerful to include whether the authors consider that these data made any difference to how these women were managed throughout their pregnancies, whether the outcome of the pregnancy itself was affected and lastly, how the authors see these data being applied in a wider clinical sense to make a difference to how we manage pregnancies in the future. As things stand with these data, I cannot see any material difference to how pregnancy is managed as a result of this work.
Author Response
Response to Reviewers’ Comments
Reviewer 1
Open Review
(x) I would not like to sign my review report
( ) I would like to sign my review report
English language and style
( ) Extensive editing of English language and style required
( ) Moderate English changes required
(x) English language and style are fine/minor spell check required
( ) I don't feel qualified to judge about the English language and style
|
Yes |
Can be improved |
Must be improved |
Not applicable |
|
|
Does the introduction provide sufficient background and include all relevant references? |
( ) |
( ) |
(x) |
( ) |
|
Is the research design appropriate? |
( ) |
( ) |
(x) |
( ) |
|
Are the methods adequately described? |
(x) |
( ) |
( ) |
( ) |
|
Are the results clearly presented? |
(x) |
( ) |
( ) |
( ) |
|
Are the conclusions supported by the results? |
( ) |
( ) |
(x) |
( ) |
Comments and Suggestions for Authors
Thyroid dysfunction among hypertensive women in Warri, Delta State Nigeria.
Nawabudike and Emokpae
I enjoyed reading this paper and commend the authors for their work and efforts to bring it together. I always approach other authors’ works with the premise that I am going to recommend publication unless there are glaring errors. I fear that this study has a few of those - to which the authors may wish to give some thought.
I understand the equation that determined the sample size and the authors have applied it accurately, but the study size (150 hypertensive pregnant women) does seem low and the numbers in the control groups (25 in each) similarly look too low to make any valid conclusions from the data. Rather than settle for the minimum number as defined by this formula, it would have been better to have increased the sample size to make any conclusions the authors draw seem more convincing.
Response:
The limitations of this study are the small sample size and the fact that it was conducted in two major towns in the State. It therefore means that generalization to the Nigeria cannot be made. Further studies involving large sample size is suggested.
It is known that thyroid function changes throughout pregnancy e.g., increased thyroid binding globulin in pregnancy will cause rises in total T3 and total T4. Increased HCG in the first trimester will suppress TSH to start with and this effect disappears by week 16 where the TSH becomes more reliable as a measure of thyroid function. I’m not sure that the authors have included details like this in their work.
Response:
Pregnancy related hormones (human chorionic gonadotropin (hCG) and estrogen can cause increase levels of T3 and T4. Human chorionic gonadotropin does stimulate the thyroid gland mimicking TSH action thereby suppressing TSH production, while estrogen stimulates the production of thyroid binding globulin (the transporter of thyroid hormones). These physiologic actions caused T3 and T4 to increase during the first trimester until about the 16 weeks of gestation and them stabilized. This can sometimes make thyroid hormone tests difficult to interpret.
I would be interested to learn how “mental illness” is defined by the authors and why it became an exclusion factor in their study population. I understand how thyroid disease can cause depressive illness, anxiety etc but I’m much less familiar with a reverse causative relationship i.e., does depressive illness, anxiety etc cause thyroid dysfunction and thus exclude such patients from this study.
Response
Thyroid disorder can affect can cause either anxiety or depression. Generally, the more severe the thyroid disease, the more severe the mood changes. Both hyperthyroidism and hypothyroidism could cause unusual nervousness such as restlessness, anxiety, irritability and depression. It is not yet clear if these symptoms are the only evidence of thyroid disease. Hence it was used as one of the exclusion criteria.
The assay principle is described in minute detail – as this is a non-commercial ELISA assay for all three analytes, this probably has some relevance, but I would suggest that most readers of this work will not be particularly interested in the finer details. However, as these results are not generated from commercial kits (and there is no criticism here), the authors really have to say whether these assays are subjected to an external QC scheme and that the performance of the assays are acceptable within that QC scheme. Otherwise, how are we meant to believe the significance of any results generated?
A set of calibrators were assayed to ensure linearity, commercial quality control sera were included in the runs and samples were determined in duplicates and mean values calculated.
Moving onto the presentation of the results: I’m sorry to say that I didn’t learn anything that I didn’t know already. However, one of my major difficulties with these data is that as far as I can see, the authors don’t define any reference ranges for their results: perhaps they haven’t defined the reference ranges for their ELISA assays but without this, it is very difficult to draw any conclusions from their work.
The locally established reference ranges are:
TSH= 0.50 – 4.1µIU/mL
T3 = 0.52 – 1.85ng/dL
T4 = 4.8 – 11.6µg/dL
The thyroid dysfunctions were defined based on the locally established reference ranges as follows:
Hypothyroidism- TSH > 4.1µIU/mL with T4<4.8µg/dL
Subclinical hypothyroidism- TSH >4.1 µIU/mL with normal T4 and T3.
Hyperthyroidism – TSH <0.5µIU/mL with T4 >11.6µg/dL
Similarly, the authors say that a number of women had overt hypothyroidism and some had subclinical hypothyroidism, but they have not defined what they consider the TSH to be in such cases. I would have said that at TSH above 10mU/L would be considered to represent overt hypothyroidism and values above the top of the reference range but <10mU/L as defining “subclinical hypothyroidism”. As far as I can see (and remember, I don’t know what is considered a normal value in their assays) all the TSH values in Tables 2 and 6 for example, look completely normal! There may be statistically significant differences between the study groups but if all these differences lie within a normal range, I wonder what significance that has i.e., as a clinician, what do I do to intervene when I see a slightly higher TSH value that still lies within my reference range? Given how labile TSH can be, I would do nothing to alter my management. In Table 6, if I’m reading this correctly, the authors are suggesting a TSH of 0.41 and TSH of 0.32 represent subclinical and overt hypothyroidism respectively. The TSH values don’t suggest any form of hypothyroidism to me and in fact these TSH values are actually lower than the euthyroid value of 2.62 – that just doesn’t make sense.
Response:
Table 6. Distribution of Thyroid Dysfunction among Hypertensive Pregnant women.
|
Thyroid function status |
Number of subjects (%) |
T3 levels (Ref. range 0.52-1.85) |
T4 levels (Ref. range 4.8-11.6) |
TSH levels (Ref range:0.5-4.1) |
|
Euthyroid |
122(81.3) |
0.98±0.21 |
6.24±1.60 |
2.62±0.62 |
|
Subclinical hypothyroidism |
15(10) |
1.01±0.21 |
11.2±0.81 |
4.81±0.19 |
|
Overt Hypothyroidism |
13(8.7) |
0.97±0.72 |
4.1±0.12 |
4.92±0.16 |
|
Total |
150(100) |
1.02±0.06 |
6.53±0.85 |
3.08±0.09 |
Reference ranges in parenthesis; TSH= Thyroid stimulating hormone; T3= triiodothyronine; T4= Thyroxine.
It would have been more powerful to include whether the authors consider that these data made any difference to how these women were managed throughout their pregnancies, whether the outcome of the pregnancy itself was affected and lastly, how the authors see these data being applied in a wider clinical sense to make a difference to how we manage pregnancies in the future. As things stand with these data, I cannot see any material difference to how pregnancy is managed as a result of this work.
Response:
Pregnant women with hypothyroidism could be at risk of obstetrical complications (including intrauterine fetal death, gestational hypertension, placentalabruption, poor perinatal outcome).Evidence has shown that thyroid hormone administration significantly improves such complications. However, babies of hypothyroid mothers may be healthy without evidence of thyroid dysfunction (Westman et al., 1995). Conversely, someauthors have reported a risk of higher perinatal mortality, congenital malformation and low birth weight(Glinoer, 1997).

Reviewer 2 Report
I have read your interesing paper. Is it weel known the relevance of thyroid functionality during the pregnancy as well as the interection between hypertensive disorders and the values of thyroid hormones.
In Materials and method you report , like anthropometic and demographic measurements,
the BMI of the patients.
In the dicussion you don’t speak about this parameter.
I think this is an important question, because the BMI can influence the presence of hypertension disorders. Can you discuss this question?
Author Response
Reviewer 2
Open Review
(x) I would not like to sign my review report
( ) I would like to sign my review report
English language and style
( ) Extensive editing of English language and style required
( ) Moderate English changes required
(x) English language and style are fine/minor spell check required
( ) I don't feel qualified to judge about the English language and style
|
Yes |
Can be improved |
Must be improved |
Not applicable |
|
|
Does the introduction provide sufficient background and include all relevant references? |
(x) |
( ) |
( ) |
( ) |
|
Is the research design appropriate? |
( ) |
(x) |
( ) |
( ) |
|
Are the methods adequately described? |
(x) |
( ) |
( ) |
( ) |
|
Are the results clearly presented? |
(x) |
( ) |
( ) |
( ) |
|
Are the conclusions supported by the results? |
(x) |
( ) |
( ) |
( ) |
Comments and Suggestions for Authors
I have read your interesing paper. Is it weel known the relevance of thyroid functionality during the pregnancy as well as the interection between hypertensive disorders and the values of thyroid hormones.
In Materials and method you report , like anthropometic and demographic measurements,
the BMI of the patients.
In the dicussion you don’t speak about this parameter.
I think this is an important question, because the BMI can influence the presence of hypertension disorders. Can you discuss this question?
Response:
The mean BMI of women hypertension was was significantly higher (p<0.001) than both pregnant women without hypertension and non-pregnant women. Obesity is associated with adverse implications on maternal and child health. Because excessive weight gain during pregnancy has been associated with risk of developing hypertension, entering pregnancy with healthier BMI has a better immediate and lon-term outcomes. Some authors have shown that the risk of gestational hypertension correlated positively with BMI with a relative risk of 2.24 for a BMI increase of 3kg/m2(Cooyannakis and Khalil, 2019).
Submission Date
28 February 2022
Date of this review
20 Mar 2022 19:26:45

Round 2
Reviewer 1 Report
I am much happier with the revised version and I am pleased to recommend its publication.